# Beyond Random Augmentations: Pretraining with Hard Views

**Fabio Ferreira**[*]
University of Freiburg

**Ivo Rapant**[*]
University of Freiburg

**Jörg K.H. Franke**
University of Freiburg

**Frank Hutter**
ELLIS Institute Tübingen &
University of Freiburg

## Abstract

Self-Supervised Learning (SSL) methods typically rely on random image augmentations, or *views*, to make models invariant to different transformations. We hypothesize that the efficacy of pretraining pipelines based on conventional random view sampling can be enhanced by explicitly selecting views that benefit the learning progress. A simple yet effective approach is to select *hard views* that yield a higher loss. In this paper, we propose *Hard View Pretraining (HVP)*, a learning-free strategy that extends random view generation by exposing models to more challenging samples during SSL pretraining. HVP encompasses the following iterative steps: 1) randomly sample multiple views and forward each view through the pretrained model, 2) create pairs of two views and compute their loss, 3) adversarially select the pair yielding the highest loss according to the current model state, and 4) perform a backward pass with the selected pair. In contrast to existing hard view literature, we are the first to demonstrate hard view pretraining's effectiveness at scale, particularly training on the full ImageNet-1k dataset, and evaluating across multiple SSL methods, ConvNets, and ViTs. As a result, HVP sets a new state-of-the-art on DINO ViT-B/16, reaching 78.8% linear evaluation accuracy (a 0.6% improvement) and consistent gains of 1% for both 100 and 300 epoch pretraining, with similar improvements across transfer tasks in DINO, SimSiam, iBOT, and SimCLR.

## 1 Introduction

Learning effective and generalizable visual representations in Self-Supervised Learning (SSL) has been approached in various ways. Many SSL methods can be broadly categorized into generative and discriminative approaches (Chen et al., 2020a). Generative methods focus on generating image inputs, while discriminative methods, particularly contrastive learning (Hadsell et al., 2006; He et al., 2020), aim at learning latent representations in which similar image views are located closely, and dissimilar ones distantly.

Such views are generated by applying a sequence of (randomly sampled) image transformations and are usually composed of geometric (cropping, rotation, etc.) and appearance (color distortion, blurring, etc.) transformations. Prior work (Chen et al., 2020a; Wu et al., 2020; Purushwalkam & Gupta, 2020; Wagner et al., 2022; Tian et al., 2020b) has identified *random resized crop* (RRC), which randomly crops the image and resizes it back to a fixed size, as well as color distortion as critical transformations for effective representation learning. However, despite this finding and to our best knowledge, little research has gone into identifying more effective ways for generating views to improve performance.

Existing SSL approaches that attempt to control the hardness of views include adversarial (Shi et al., 2022; Tamkin et al., 2021) or cooperative (Hou et al., 2023) techniques. For

---

[*]Equal contribution. Correspondence to: ferreira@cs.uni-freiburg.de

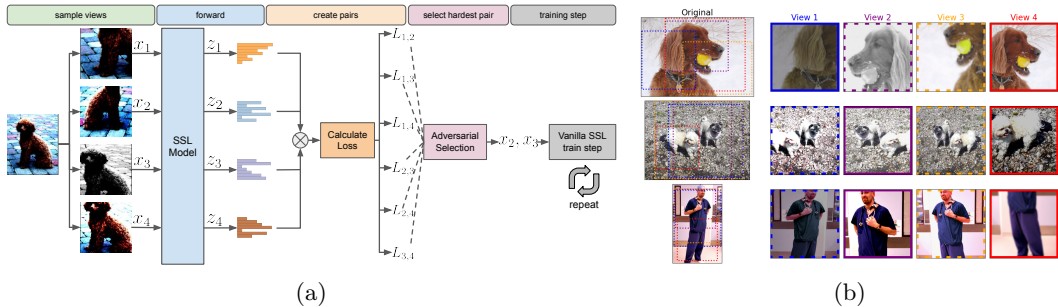

(a)               (b)

Figure 1: **(a)** HVP first samples $N$ views, pairs them, and adversarially selects the hardest pair, i.e., the one with the worst loss according to the current model state. **(b)** Examples (left) and sampled views (right) after transformations. Hard pairs selected by HVP are shown with a solid frame.

instance, Tian et al. (2020b) use mutual information theory to adversarially learn view generators. These methods offer valuable insights into optimizing views but often introduce additional complexity, such as requiring additional models or significant changes to the SSL pipeline, limiting their practicality in state-of-the-art models where resource constraints are already a concern. Similar to what we propose, Koçyigit et al. (2023) introduce a hard view sampling strategy but targeting the acceleration of pretraining. Moreover, their approach requires tuning hyperparameters like learning rate and augmentation magnitude, which can introduce confounding variables. Furthermore, neither Koçyigit et al. (2023) nor Tian et al. (2020b) validate their methods on larger datasets such as ImageNet-1k (Deng et al., 2009), or across larger model architectures, limiting their scalability and applicability.

Building on these observations, we propose *Hard View Pretraining (HVP)*, a fully learning-free, easy-to-integrate approach designed to improve standard pretraining methods without the need for additional model training or complex modifications. Our method leverages the current model state to select challenging samples during pretraining by adversarially sampling pairs of views and selecting the pair that yields the highest loss according to the model's current state (see Fig. 1a). Unlike previous approaches, HVP requires no hyperparameter tuning and demonstrates scalability to large datasets like ImageNet-1k, offering an efficient and practical solution for improving SSL pretraining. To the best of our knowledge, we are the first to demonstrate the effectiveness of a hard view sampling strategy at scale, particularly on modern architectures like Vision Transformers (ViTs) and training on the full ImageNet dataset. Our approach not only integrates seamlessly with recent state-of-the-art SSL methods but also showcases consistent improvements across both convolutional architectures and ViTs, validating its robustness and scalability.

Overall, our contributions can be summarized as follows:

- We propose *Hard View Pretraining (HVP)*, an easy-to-use method complementing SSL by extending the common random view generation to automatically expose the model to harder samples during pretraining. HVP simply requires the ability to compute sample-wise losses;

- We demonstrate the effectiveness and compatibility of our approach using ImageNet-1k pretraining across four popular SSL methods that cover a diverse range of discriminative objectives such as SimSiam (Chen & He, 2021), DINO (Caron et al., 2021), iBOT (Zhou et al., 2021), and SimCLR (Chen et al., 2020a);

- HVP achieves a new state-of-the-art result on DINO ViT-B/16, improving over the officially reported baseline of 78.2% linear evaluation accuracy by reaching 78.8% (400 epochs). HVP also consistently improves all other baselines by an average of 1% in linear evaluation on ImageNet across 100 and 300 epoch-pretraining runs;

- We show similar improvements on a diverse set of transfer tasks, including finetuning, object detection, and segmentation, and present insights into the underlying mechanisms and robustness of HVP.

We make our PyTorch Paszke et al. (2019) code, models, and all used hyperparameters publicly available under `https://github.com/automl/hvp`.

## 2 Related Work

### 2.1 Discriminative Self-Supervised Learning

The core idea behind the discriminative learning framework (Chen et al., 2020a) is to learn image representations by contrasting positive pairs (two views of the same image) against negative pairs (two views of different images) (Hadsell et al., 2006). To work well in practice and to prevent model collapse, contrastive learning methods often require a large number of negative samples (Wu et al., 2018; van den Oord et al., 2018; Chen et al., 2020a; He et al., 2020; Tian et al., 2020a; Chen et al., 2020b) stored in memory banks (Wu et al., 2018; He et al., 2020) or, for instance, in the case of SimCLR, implicitly in large batches (Chen et al., 2020a). Non-contrastive approaches, such as BYOL (Grill et al., 2020), SimSiam (Chen & He, 2021), DINO (Caron et al., 2021) and others (Zbontar et al., 2021; Caron et al., 2020; Ermolov et al., 2021), can only use positive pairs without causing model collapse but rely on other techniques, such as Siamese architectures, whitening of embeddings, clustering, maximizing the entropy of the embeddings, momentum encoders, and more.

### 2.2 Optimizing for Hard Views in SSL

Due to its performance-improving benefits, the realm of learning task-specific augmentation policies based on data has seen quick development (Cubuk et al., 2019; Ho et al., 2019; Lin et al., 2019; Zhang et al., 2020; Hataya et al., 2020; Hou et al., 2023; Müller & Hutter, 2021). However, these approaches do not include the random resize crop operation in their search spaces, limiting the control of view hardness. Similar to us, Koçyigit et al. (2023) uses the current model state for selecting hard views. However, their approach requires controlling learning hyperparameters, while mostly training on a smaller version of ImageNet and ResNets (He et al., 2016) only. We offer a more complete analysis of hard view pretraining, demonstrating the benefits without potential confounding factors such as hyperparameter adjustments. We also focus on performance rather than pretraining speedups and employ higher budgets (longer pretraining, larger batch sizes) on the full ImageNet dataset and both ResNets and ViTs (Dosovitskiy et al., 2020)). Other works utilize additional networks for view generation, such as Tamkin et al. (2021); Shi et al. (2022); Tian et al. (2020b) (adversarial view generators), Peng et al. (2022) (localization network for semantic awareness), Li et al. (2024) (pretrained generative models to enhance augmentation quality), and Han et al. (2023) (leveraging a pretrained GAN in a SimCLR-only setting). However, unlike HVP, all these methods add non-trivial complexity to the training pipeline by requiring learning auxiliary and adversarial components or are often limited in their applicability across different SSL frameworks (e.g., by requiring negative view pairs).

## 3 Method

### 3.1 Self-supervised Learning Framework

In this section, we introduce our approach, which is also depicted in Algorithm 1. Many different self-supervised discriminative learning (He et al., 2020) objectives exist, each characterized by variations stemming from design choices, such as by the use of positive and negative samples or asymmetry in the encoder-projector network structure. For simplicity of exposure, we will introduce our approach based on the SimSiam objective (Chen & He, 2021), but we do note that our method can be used with any other discriminative SSL objective that allows the computation of sample-wise losses.

SimSiam works as follows. Assume a given set of images $\mathcal{D}$, an image augmentation distribution $\mathcal{T}$, a minibatch of $M$ images $\mathbf{x} = \{x_i\}_{i=1}^{M}$ sampled uniformly from $\mathcal{D}$, and two sets of randomly sampled image augmentations $A$ and $B$ sampled from $\mathcal{T}$. SimSiam applies

$A$ and $B$ to each image in $\mathbf{x}$ resulting in $\mathbf{x}^A$ and $\mathbf{x}^B$. Both augmented sets of views are subsequently projected into an embedding space with $\mathbf{z}^A = g_\theta(f_\theta(\mathbf{x}^A))$ and $\mathbf{h}^B = f_\theta(\mathbf{x}^B)$ where $f_\theta$ represents an encoder (or backbone) and $g_\theta$ a projector network. SimSiam then minimizes the following objective:

$$\mathcal{L}(\theta) = \frac{1}{2}\left(D(\mathbf{z}^A, \mathbf{h}^B) + D(\mathbf{z}^B, \mathbf{h}^A)\right) \tag{1}$$

where $D$ denotes the negative cosine similarity function. Intuitively, when optimizing $\theta$, the embeddings of the two augmented views are attracted to each other.

## 3.2 Pretraining with Hard Views

We now formalize how we expose the model to more challenging views during pretraining. In a nutshell, Hard View Pretraining extends the random view generation by sampling adversarially harder views during pretraining. Instead of having two sets of augmentations $A$ and $B$, we now sample $N$ sets of augmentations, denoted as $\mathcal{A} = \{A_1, A_2, \ldots, A_N\}$. Each set $A_i$ is sampled from the image augmentation distribution $\mathcal{T}$, and applied to each image in $\mathbf{x}$, resulting in $N$ augmented sets of views $\mathbf{x}^{A_1}, \mathbf{x}^{A_2}, \ldots, \mathbf{x}^{A_N}$. Similarly, we obtain $N$ sets of embeddings $\mathbf{z}^{A_1}, \mathbf{z}^{A_2}, \ldots, \mathbf{z}^{A_N}$ and predictions $\mathbf{h}^{A_1}, \mathbf{h}^{A_2}, \ldots, \mathbf{h}^{A_N}$ through the encoder and projector networks. We then define a new objective function that seeks to find the pair $(x_i^{A_k}, x_i^{A_l})$ of a given image $x_i$ that yields the highest loss:

$$\begin{aligned}
(x_i^{A_k}, x_i^{A_l}) &= \arg\max_{k,l;k\neq l} \mathcal{L}(\theta)_{i,k,l} \\
&= \arg\max_{k,l;k\neq l} \frac{1}{2}\left(D(z_i^{A_k}, h_i^{A_l}) + D(z_i^{A_l}, h_i^{A_k})\right),
\end{aligned} \tag{2}$$

where $\mathcal{L}(\theta)_{i,k,l}$ simply denotes a sample-wise variant of Eq. 1.

Overall, we first generate $N$ augmented views for each image $x_i$ in the minibatch. Then, we forward these augmented views through the networks and create all combinatorially possible $\binom{N}{2}$ pairs of augmented images. Subsequently, we use Eq. 2 to compute the sample-wise loss for each pair. We then select all pairs that yielded the highest loss to form the new *hard* minibatch of augmented sets $\mathbf{x}^{A_{k*}}$ and $\mathbf{x}^{B_{l*}}$, discard the other pairs and use the hard minibatch for optimization. As shown in Algorithm 1, we repeat this process in each training iteration.

---

**Algorithm 1** Pretraining with Hard Views

1: **Input:** Number of views $N \geq 2$, batch size $M$,
2: augmentation distribution $\mathcal{T}$, model $f$
3: **for** each $x_i$ in the sampled batch $\{x_i\}_{i=0}^M$ **do**
4:     Sample $N$ augmentations: $A = \{t_n \sim \mathcal{T}\}_{n=0}^N$
5:     Create augmented views: $\mathbf{x}_i^A = \{t_n(x_i)\}_{n=0}^N$
6:     Forward all views through $f$
7:     Create all $\binom{N}{2}$ view pairs $\mathbf{x}_i^{A_k} \times \mathbf{x}_i^{A_l}$, $k \neq l$
8:     Add *hard* pair $(x_i^{A_{k*}}, x_i^{A_{l*}})$ that maximizes Eq. 2 to the new batch with only hard pairs
9: Proceed with standard SSL training
10: Repeat for all batches
11: **return** Pretrained model $f$

---

Intuitively, our approach introduces a more challenging learning scenario in which the model is encouraged to learn more discriminate features by being exposed to harder views. In the early stage of training, the embedding space lacks a defined structure for representing similarity among views. As training progresses, our method refines the concept of similarity through exposure to views that, from the perspective of the model, remain challenging given its current state. By limiting the number of sampled views, we upper-bound the difficulty of learning to prevent tasks from becoming too difficult and hindering learning. This ensures a controlled evolution of the embedding space, where the model's perception of difficulty is continuously challenged in tandem with its growing capacity to differentiate views. Consequently, HVP can be seen as a regularization that prevents the model from overfitting to easy views.

While we exemplified the integration of HVP with the SimSiam objective, integrating it into other contrastive methods is as straightforward. The only requirement of HVP is to

be able to compute sample-wise losses (to select the views with the highest loss). In our experiments section and in addition to SimSiam, we study the application of HVP to the objectives of DINO, iBOT, and SimCLR (see also Appendix K.1 for a formal exemplary definition for the integration of HVP into SimCLR).

## 4 Implementation and Evaluation Protocols

### 4.1 Implementation

We now describe the technical details of our approach. HVP can be used with any SSL method that allows computing sample-wise losses, and the only two elements in the pipeline we adapt are: 1) the data loader (which now needs to sample $N$ views for each image) and 2) the forward pass (which now invokes a *select* function to identify and return the hard views). The image transformation distribution $\mathcal{T}$ taken from the baselines is left unchanged. Note, for the view selection one could simply resort to random resized crop (RRC) only and apply the rest of the operations after the hard view selection (see Section 6.1 for a study on the influence of appearance on the selection).

All experiments were conducted with $N = 4$ sampled views, yielding $\binom{N}{2} = 6$ pairs to compare, except for DINO which uses 10 views (2 global, 8 local heads) per default. For DINO, we apply HVP to both global and local heads but to remain tractable, we upper-bound the number of total pair comparisons to 128. SimCLR uses positive and negative samples. Following the simplicity of HVP, we do not alter its objective, which naturally leads to selecting hard views that are adversarial to positive and "cooperative" to negative views. For iBOT, we use the original objective as defined by the authors with global views only.

### 4.2 Evaluation Protocols

We now describe the protocols used to evaluate the performance in our main results section. In self-supervised learning, it is common to assess pretraining performance with the linear evaluation protocol by training a linear classifier on top of frozen features or finetuning the features on downstream tasks. Our general procedure is to follow the baseline methods as closely as possible, including hyperparameters and code bases (if reported). It is common to use RRC and horizontal flips during training and report the test accuracy on central crops. Due to the sensitivity of hyperparameters, and as done by Caron et al. (2021), we also report the quality of features with a simple weighted nearest neighbor classifier (k-NN).

| Method | Arch. | 100 epochs | | 300 epochs | |
|---|---|---|---|---|---|
| | | Lin. | $k$-NN | Lin. | $k$-NN |
| DINO | ViT-S/16 | 73.52 | 68.80 | 75.48 | 72.62 |
| + HVP | ViT-S/16 | 74.67 | 70.72 | 76.56 | 73.65 |
| **Impr.** | | **+1.15** | **+1.92** | **+1.08** | **+1.03** |
| DINO | RN50 | 71.93 | 66.28 | 75.25 | 69.53 |
| + HVP | RN50 | 72.87 | 67.33 | 75.65 | 70.05 |
| | | **+0.94** | **+1.05** | **+0.40** | **+0.52** |
| SimSiam | RN50 | 68.20 | 57.47 | 70.35 | 61.40 |
| + HVP | RN50 | 68.98 | 58.97 | 70.90 | 62.97 |
| | | **+0.78** | **+1.50** | **+0.55** | **+1.57** |
| SimCLR | RN50 | 63.37 | 52.83 | 65.50 | 55.65 |
| + HVP | RN50 | 65.33 | 54.76 | 67.30 | 56.80 |
| | | **+1.96** | **+1.93** | **+1.80** | **+1.15** |
| iBOT | ViT-S/16 | 69.55 | 62.93 | 72.76 | 66.92 |
| + HVP | ViT-S/16 | 70.27 | 62.75 | 73.99 | 67.16 |
| | | **+0.73** | **-0.18** | **+1.23** | **+0.24** |

Table 1: Average top-1 linear and $k$-NN classification accuracy on the ImageNet validation set for 100 and 300-epoch pretrainings across 3 seeds.

## 5 Main Results

Here, we discuss our main results on image classification, object detection, and segmentation tasks. All results are self-reproduced using the original baseline code and hyperparameters (see Appendix Section A for details).

### 5.1 Evaluations on ImageNet

We report the top-1 validation accuracy on frozen features, as well as the k-NN classifier performance, in Table 1. For DINO, we additionally compare ResNet-50 (He et al., 2016)

| Method | Arch. | CIFAR10 | | CIFAR100 | | Flowers102 | | iNat 21 | | Food101 | |
|---|---|---|---|---|---|---|---|---|---|---|---|
| | | Lin. | F.T. | Lin. | F.T. | Lin. | F.T. | Lin. | F.T. | Lin. | F.T. |
| SimSiam | RN50 | 82.60 | 95.50 | 54.20 | 77.20 | 34.27 | 56.40 | 32.50 | 60.30 | 65.70 | 83.90 |
| + HVP | RN50 | 84.40 | 96.10 | 57.10 | 78.20 | 38.37 | 58.90 | 33.90 | 60.90 | 67.10 | 84.70 |
| **Impr.** | | **+1.80** | **+0.60** | **+2.90** | **+1.00** | **+4.10** | **+2.50** | **+1.40** | **+0.60** | **+1.40** | **+0.80** |
| DINO | ViT-S/16 | 94.53 | 98.53 | 80.63 | 87.90 | 91.10 | 93.20 | 46.93 | 53.97 | 73.30 | 87.50 |
| + HVP | ViT-S/16 | 95.13 | 98.65 | 81.27 | 88.23 | 92.07 | 93.60 | 49.03 | 54.16 | 74.13 | 87.91 |
| **Impr.** | | **+0.60** | **+0.12** | **+0.63** | **+0.33** | **+0.97** | **+0.40** | **+2.10** | **+0.19** | **+0.83** | **+0.41** |

Table 2: HVP compares favorably against models trained without it when fine-tuned (F.T.) to or linearly evaluated (Lin.) on other datasets (averaged over 3 seeds; 100-ep. preraining).

against the ViT-S/16 (Dosovitskiy et al., 2020) architecture. We point out that both methods, vanilla, and vanilla+HVP always receive the same number of data samples for training. Our method compares favorably against all baselines with an increased performance of approximately 1% on average for 100 and 300 epoch pretraining, showing the benefit of sampling hard views.

Due to limited computing resources, we run the majority of pretrainings in this paper for 100 epochs and 300 epochs (200 epochs for SimCLR) and batch sizes of 512 (100 epoch) or 1024 (200 & 300 epoch trainings), respectively. This choice is in line with a strategy that favors the evaluation of a diverse and larger set of baselines over the evaluation of a less diverse and smaller set and underpins the broad applicability of HVP. We primarily ran our experiments with 8xNVIDIA GeForce RTX 2080 Ti nodes, with which the pretraining and linear evaluation duration ranged from ∼3.5 to ∼25 days. While HVP requires roughly 2x the training time of the DINO baseline (see Appendix J for further discussion on the time complexity of HVP), this investment translates directly into improved performance.

Rather than prioritizing efficiency, our focus was on achieving state-of-the-art results, highlighting HVP's flexibility and robustness across training durations. To showcase this point, we explored the scalability of HVP with larger models and extended training schedules and achieved a new **state-of-the-art result of 78.8% accuracy in linear evaluation, improving over the officially reported baseline of 78.2% on DINO ViT-B/16 (400 epochs)**. For k-NN classification, the same model similarly surpassed the DINO baseline by 0.85%, reaching 76.95% compared to 76.1%. These results demonstrate the scalability of our method to larger models and longer pretraining schedules. We emphasize that HVP is insensitive to the baseline hyperparameters and simply reusing the default ones consistently resulted in improvements in the reported magnitudes across all experiments.

## 5.2 Transfer to Other Datasets and Tasks

We now report the transferability of features learned with HVP. For all experiments here, we use our 100-epoch ImageNet pretrained iBOT and DINO ViT-S/16 models, respectively.

### 5.2.1 Linear Evaluation and Finetuning

In Table 2, we apply both the linear evaluation (Lin.) and finetuning (F.T.) protocols to our models across a diverse set of datasets consisting of CIFAR10 (Krizhevsky, 2009), CIFAR100, Flowers102 (Nilsback & Zisserman, 2008), Food101 (Bossard et al., 2014), and iNaturalist 2021 (iNaturalist 2021 competition dataset). Our results show that the improvements achieved by sampling hard views that we observed so far also transfer to other datasets.

| Method | Arch. | OD | | IS | |
|---|---|---|---|---|---|
| | | 100 | 300 | 100 | 300 |
| iBOT | ViT-S/16 | 66.13 | 66.80 | 63.10 | 63.63 |
| + HVP | ViT-S/16 | 66.50 | 67.13 | 63.50 | 64.23 |
| **Impr.** | | **+0.37** | **+0.33** | **+0.40** | **+0.60** |
| DINO | ViT-S/16 | 65.90 | 66.60 | 62.83 | 63.63 |
| + HVP | ViT-S/16 | 66.37 | 67.00 | 63.37 | 64.00 |
| **Impr.** | | **+0.47** | **+0.40** | **+0.53** | **+0.37** |

Table 3: Object Detection (OD) and Instance Segmentation (IS) AP50 performance on COCO for 100/300 epoch pretraining.

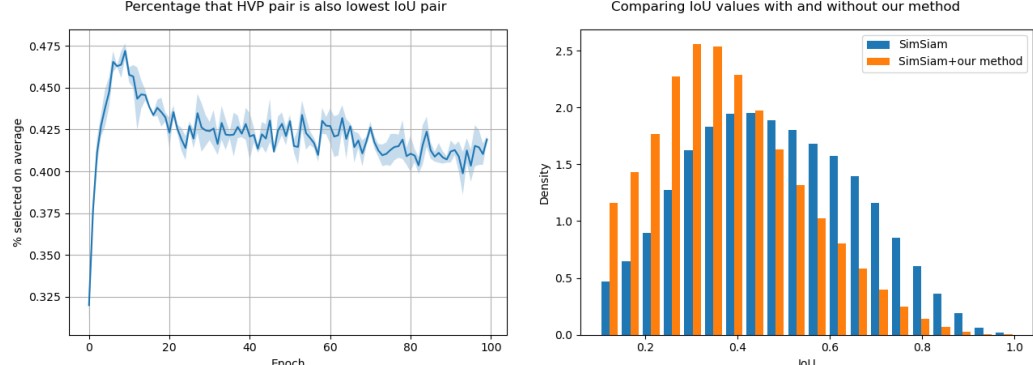

Figure 2: **Left:** In over 40% of the cases, the adversarially selected view pair has also the lowest Intersection over Union throughout SimSiam+HVP pretraining. We attribute the early spike to the random initialization of the embedding. **Right:** HVP (blue) shows a shift to smaller IoU values over standard pretraining (orange). Both results are based on 3 seeds.

### 5.2.2 OBJECT DETECTION AND INSTANCE SEGMENTATION

For object detection and instance segmentation, we use the COCO (Lin et al., 2014) dataset with Cascade Mask R-CNN (Cai & Vasconcelos, 2019; He et al., 2017). Table 3, where we report the AP50 performance, shows that the features learned with HVP also transfer favorably to these tasks and outperform the iBOT and DINO baseline with a 100-ep. and 300-ep. pretraining. More details and performance results on this task are provided in Appendix H.1.

## 6 EMPIRICAL ANALYSIS OF HVP

In this section, we discuss studies designed to shed light on the mechanisms underlying HVP. We address the following questions: 1) "Which pattern can be observed that underlies the hard view selection?" and 2) "What are the effects on empowering the adversary?". For all experiments conducted here, we use our 100-epoch, ImageNet-pretrained SimSiam+HVP models with four sampled views. In Appendix F.1, we also analyze whether we can infer a "manual" augmentation policy from the following observed patterns.

### 6.1 Q1: WHICH PATTERNS CAN BE OBSERVED WITH HVP?

When visually studying examples and the views selected by HVP in Figures 1b and 5 (in the appendix), we notice that both geometric and appearance characteristics seem to be exploited, for instance, see the brightness difference between the views of the first two rows in Fig. 1b. We also see a generally higher training loss (Fig. 6 in the appendix) indicative of an increased task difficulty.

### 6.1.1 LOGGING AUGMENTATION DATA

To assess these observations, we logged relevant hyperparameter data during SimSiam training with HVP. The logs include for each view the sampled geometric and appearance parameters from the data augmentation operations (such as the height/width of the crops or the brightness; see Section E in the appendix for more details), as well as the loss and whether the view was selected. As evaluated metrics, we chose the Intersection over Union (IoU), Relative Distance (normalized distance of the center points views), color distortion distance (the Euclidean distance between all four color distortion parameters), and the individual color distortion parameters.

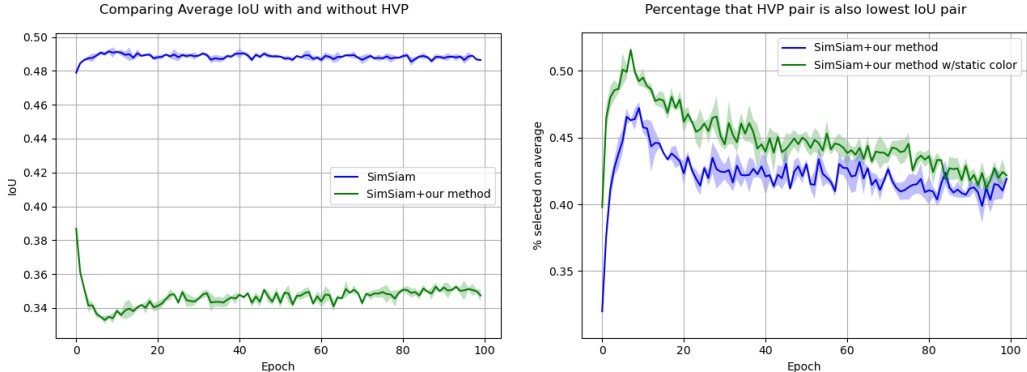

Figure 3: **Left:** The average IoU of view pairs selected by SimSiam+HVP (blue) compared against the default SimSiam training (green). **Right:** Using static color augmentation for all pairs before the selection increases the dependency on the IoU.

### 6.1.2 Importance of Augmentation Metrics

Given 300k such samples, we then used fANOVA (Hutter et al., 2014) to determine how predictive these metrics are. This resulted in the metric with the highest predictive capacity on the loss being the IoU, explaining 15% of the variance in performance, followed by brightness with 5% (for more details see Fig. 8 in the appendix). The importance of IoU in HVP is further underpinned by the following observation: the fraction of view pairs selected by HVP, which are also the pairs with the lowest IoU among all six pairs (N=4), is over 40% (random: ~16.7%) during training. Moreover, when using HVP, a shift to smaller IoU values can be observed when comparing against standard pretraining (see Fig. 2).

### 6.1.3 Taking a Closer Look at the Intersection over Union

We also examined the IoU value over the course of training in Fig. 3 (left). An observable pattern is that the IoU value with HVP (Fig. 3 (left) in green) is smaller and varies more when compared against training without HVP (blue). We believe this is due to the sample-wise and stateful nature of the adversarial selection as HVP chooses different IoU values between varying samples and model states.

Lastly, we assessed the effect of the color augmentation on the pair selection. For this study, we sampled *one* set of color augmentations (as opposed to one for each view) per iteration and applied it to all views. We apply sampled data augmentations to each view only after identifying the hardest pair. As we show in Fig. 3 (right), the fraction of selected pairs that are also the hardest pairs slightly increases in this case. One possible explanation for this is that it reflects the non-negligible role of color variation between views (as shown previously with the importance analysis), where HVP is given less leverage to increase hardness through a static appearance and instead, depends more on leveraging the IoU. Another key observation is that HVP often chooses view pairs that incorporate zooming in and out or an increased distance between the views (see last row of Fig. 1b).

## 6.2 Q2: What are the Effects of Strong Adversaries?

It is well known that adversarial learning can suffer from algorithmic instability (Xing et al., 2021), e.g. by giving an adversary too much capacity. Here, we further explore the space of adversarial capacity for pretraining with hard views by adapting and varying HVP hyperparameters in order to gain a better understanding of their impact and robustness on discriminative learning. Additionally, we report further results on learning an adversary in Appendix G.1.

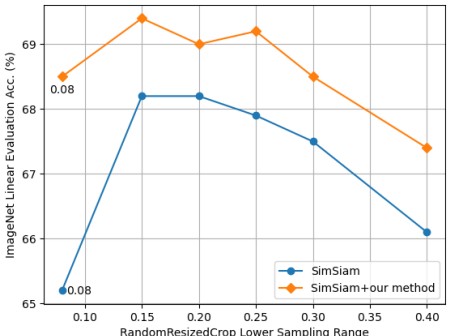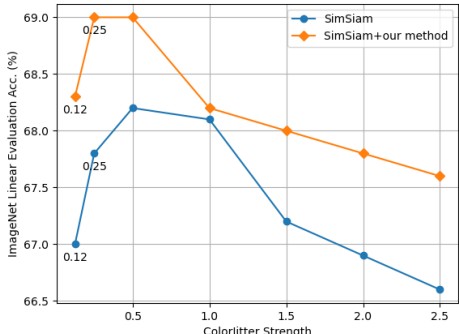

Figure 4: With HVP, SimSiam appears more robust to augmentation hyperparameter variation. We show this for RandomResizedCrop (**left**) and ColorJitter (**right**). For RRC, the values indicate the lower value of the sampling range and for CJ the intensity of the color cues. Results averaged over two seeds and SimSiam defaults are RRC=0.2 and CJ=0.5.

### 6.2.1 ROBUSTNESS TO AUGMENTATION HYPERPARAMETERS

In assessing the robustness of HVP to augmentation hyperparameters, we conducted an analysis focusing on two primary augmentation operations: RandomResizedCrop (RRC) and ColorJitter (CJ). Our findings depicted in Fig. 4 suggest that HVP enhances the robustness of SSL methods, like SimSiam, against variations in these hyperparameters. Note, when varying one operation, either RRC or CJ, we maintained the other operation at its default configuration. In both settings, we observe less performance degradation with extreme augmentation values and overall smaller degradation rates for HVP. We believe that this robustness stems from hard view pretraining which inadvertently equips the model to handle stronger augmentations.

### 6.2.2 INCREASING THE NUMBER OF VIEWS

In our initial experiments, we explored variations in the number of sampled views $N$ with SimSiam and HVP. As can be seen in Fig. 7 in the appendix, while $N = 8$ views still outperform the baseline in terms of linear evaluation accuracy, it is slightly worse than using $N = 4$ views (-0.05% for 100 epochs pretraining on linear evaluation). We interpret this result as the existence of a "sweet spot" in setting the number of views, where, in the limit, a higher number of views corresponds to approximating a powerful adversarial learner, capable of choosing very hard and unfavorable learning tasks that lead to model collapse and performance deterioration. We experimented with such an adversarial learner and report results in Appendix G.1.

## 7 CONCLUSION

We presented HVP, a new data augmentation and learning strategy for Self-Supervised Learning designed to challenge pretrained models with harder samples. This straightforward method allows pushing the effectiveness of the traditional random view generation in SSL. When combined with methods like DINO, SimSiam, iBOT, and SimCLR, HVP consistently showcased improvements of 1% on average in linear evaluation and a diverse set of transfer tasks. HVP achieved a new state-of-the-art result of 78.8% linear evaluation accuracy on DINO ViT-B/16, a 0.6% improvement over the previous baseline. This illustrates the scalability and effectiveness of our approach for larger pretraining settings. With growing models, there is an increasing demand for more data to effectively train them. Synthetic data generation offers a viable solution by enhancing the quantity and diversity of training data. Data augmentation techniques like HVP play a crucial role in this process by creating challenging views, which can serve as synthetic data. All in all, HVP holds promise for scenarios where one seeks to push absolute performance or explore making models less sensitive to hyperparameters, thereby strengthening them for various downstream applications.

## 8 ACKNOWLEDGEMENTS

Frank Hutter acknowledges the financial support of the Hector Foundation. We also acknowledge funding by the European Union (via ERC Consolidator Grant DeepLearning 2.0, grant no. 101045765). Views and opinions expressed are however those of the author(s) only and do not necessarily reflect those of the European Union or the European Research Council. Neither the European Union nor the granting authority can be held responsible for them. Moreover, we acknowledge funding by the Deutsche Forschungsgemeinschaft (DFG, German Research Foundation) under grant number 417962828, by the state of Baden-Württemberg through bwHPC and the German Research Foundation (DFG) through grant INST 35/1597-1 FUGG, as well as the Gauss Center for Supercomputing eV (www.gauss-centre.eu) for funding this project by providing computing time on the GCS supercomputer JUWELS at Jülich Supercomputing Center (JSC). We also acknowledge Robert Bosch GmbH for financial support and the computing time made available on the high-performance computer NHR@KIT Compute Cluster at the NHR Center NHR@KIT.

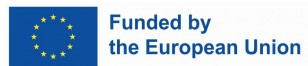

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

## Appendix

## A  Reproducibility Statement

We provide complete code, environment installation instructions, hyperparameter settings and model checkpoints here: `https://anonymous.4open.science/r/pretraining-hard-views/`. For transparency, we outline all hyperparameters, data splits, and evaluation protocols in detail in Section I. Most experiments were run across multiple seeds, and we report average results to account for variability. Information regarding the required computational resources is discussed in Section J below.

## B  Examples Sampled by HVP

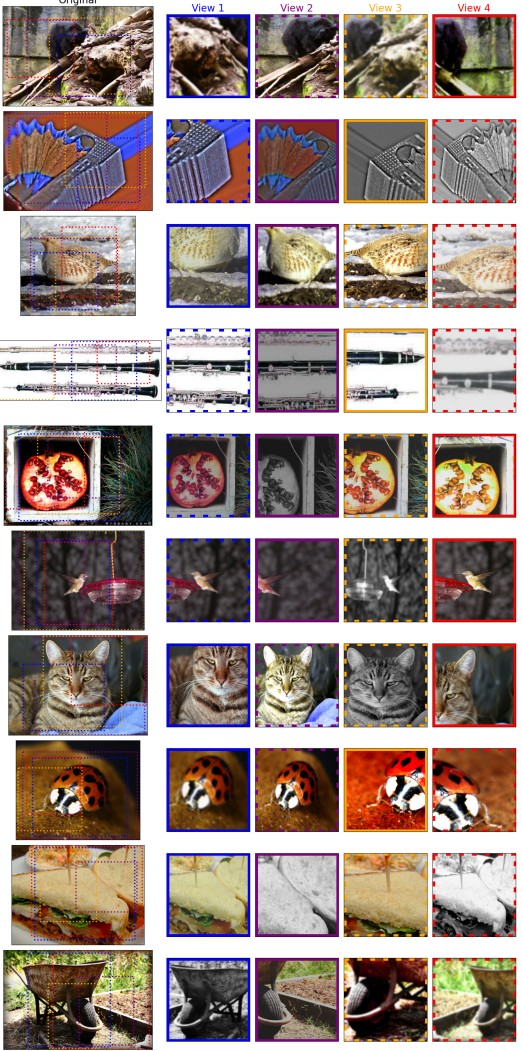

Figure 5: We depict row-wise ten example images from the ImageNet train set along with their sampled views with a finished, 100-epoch trained SimSiam ResNet50 model. Left: original image with the overlaid randomly sampled crops (colored dashed rectangles). Right: All views after applying resizing and appearance augmentations. The pair that is selected adversarially by HVP is highlighted in solid lines, eg. View 1 and View 4 in the first row.

## C  Training Loss

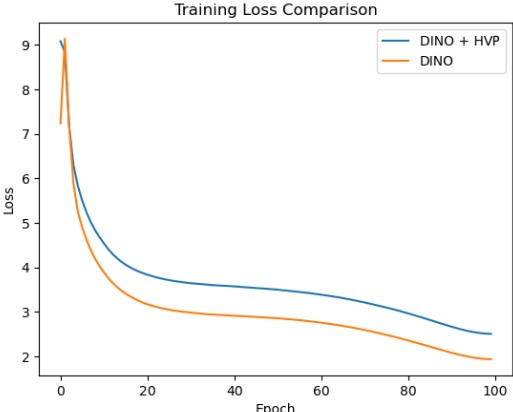

Figure 6: The training loss over 100 epochs. Comparing the DINO vanilla method with DINO + HVP. The spike and drop in the loss curve of DINO is caused by freezing the last layer in the first epoch which was proposed by the authors as a strategy to enhance downstream performance. For HVP we can only see a drop and no spike. We believe this is because HVP exposes the model to hard views from the beginning of training (i.e. the loss is immediately maximized).

## D  Effect of More Views on Linear Evaluation Performance

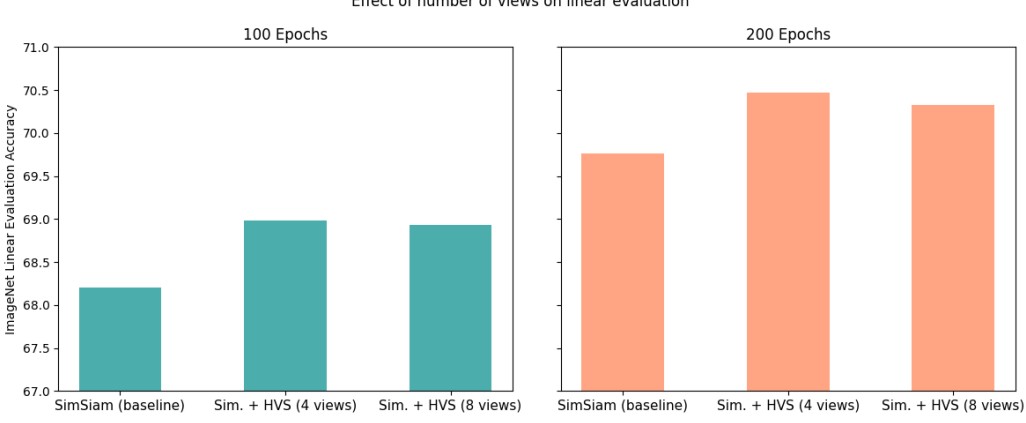

Figure 7: Setting the number of views too high can result in performance deterioration. This shows that diminishing returns exist, likely because the adversary becomes too strong, resulting in a too hard learning task.

## E  Assessing the Importance of Metrics with fANOVA

To assess the importance of various metrics on the training loss, we apply fANOVA Hutter et al. (2014) on data that we logged during training with HVP. We used 300k samples that contain the following sampled parameters from the geometric and appearance data augmentation operations for each view: all random resized crop parameters (height and width of the original image, coordinates of crop corners and height and width of the crop), all Colorjitter (color distortion) strengths (brightness, contrast, saturation, hue), grayscale on/off, Gaussian blurring on/off, horizontal flip on/off, loss, and if the crop was selected or

| IoU Policy Type | SimSiam | DINO |
|---|---|---|
| Baseline (B) | 68.20 | 73.50 |
| B+range(0.3-0.35) | -0.80 | -1.47 |
| B+range(0.3-0.35)+alt. | +0.10 | -0.45 |
| B+range(0.4-0.6) | +0.55 | -0.40 |
| B+range(0.4-0.6)+alt. | +0.25 | -0.20 |
| B+range(0.1-0.8) | -33.95 | -1.50 |
| B+range(1.0-0.1) | +0.07 | - |

Table 4: Top-1 lin. eval. accuracies for the manual IoU policy.

not. The metrics we chose are Intersection over Union (IoU), Relative Distance (sample-wise normalized distance of the center points of crop pairs), color distortion distance (the Euclidean distance between all four color distortion operation parameters, i.e. brightness, contrast, saturation, hue), and the individual color distortion parameters of the Colorjitter operation. As can be seen in Fig. 8, the metric with the highest predictive capacity on the loss is the IoU with an importance of 15.2% followed by brightness with 5.1%. The relative distance has an importance of 3.3%, the Colorjitter distance 2.3%, the contrast 1.6%, the saturation 1.4%, hue 0.6%, and all parameters jointly 1.7%.

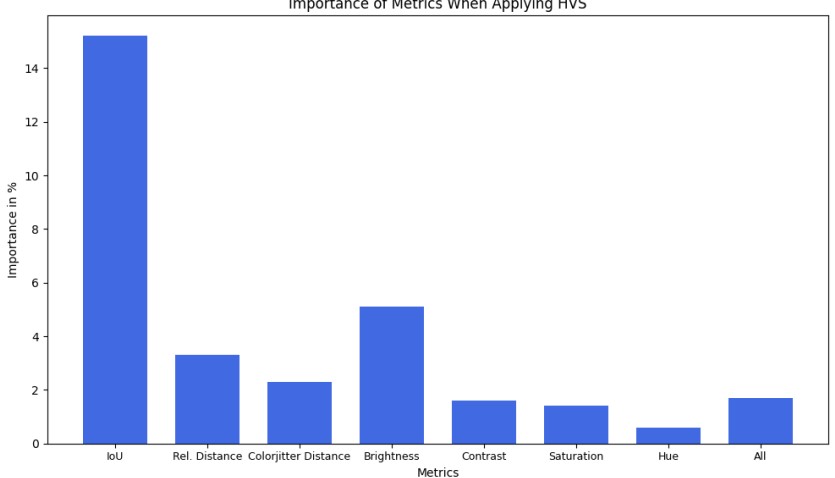

Figure 8: Application of fANOVA Hutter et al. (2014) on logged training data to determine metrics with high predictive capacity on the training loss.

## F  ADDITIONAL EMPIRICAL ANALYSIS

### F.1  CAN A MANUAL AUGM. POLICY BE DERIVED?

Since harder pretraining tasks seem beneficial according to observations made in Q1, a natural question arises: can we mimic the adversarial selection with a manually scripted augmentation policy? Such a policy would replace HVP and lower the computational cost. Since the IoU plays an important role, below, we study several ways to construct a simple manual augmentation policy based on IoU.

### F.1.1  DERIVING AN AUGMENTATION POLICY

We implemented the following rejection sampling algorithm in the augmentation pipeline: we linearly approximate the IoU values from Fig. 3 (left; in blue) with start and end values of 0.30 and 0.35 (ignoring the dip in the early phase). For each iteration, we then check if the pairs exceed the IoU value and if so, we reject the pair and re-sample a new pair. This ensures that only pairs are sampled that entail a minimum task difficulty (by means

of a small enough IoU). We varied different hyperparameters, e.g., IoU start/end ranges, inverse schedules, and alternating between the IoU schedule and the standard augmentation . Training both SimSiam and DINO models for 100 epochs yielded performance drops or insignificant improvements (see Table 4). These results indicate that using a manual policy based on metrics in pixel space such as the IoU is non-trivial. Additionally, these results show that transferring such a policy from SimSiam to DINO does not work well, possibly due to additional variations in the augmentation pipeline such as multi-crop. In contrast, we believe that HVP is effective and transfers well since it 1) operates on a similarity level of latent embeddings that may be decorrelated from the pixel space and 2) has access to the current model state.

### F.1.2 Assessing the Difficulty of Predicting the Pair Selected by HVP

We further validate the previous result and assess an upper limit on the performance for predicting the hardest pair. By fitting a gradient boosting classification tree Friedman (2001), we predict the selected view pair conditioned on all SimSiam hyperparameter log data from Q1 except for the flag that indicates whether a view was selected. As training and test data, we used the logs from two seeds (each 300k samples) and the logs from a third seed, respectively. We also tuned hyperparameters on train/valid splits and applied a 5-fold CV. However, the resulting average test performance in all scenarios never exceeded 40%, indicating that it is indeed challenging to predict the hardness of views based on parameter-level data. This further supports our hypothesis that deriving a policy for controlling and increasing hardness based on geometric and appearance parameters is non-trivial and that such a policy must function on a per-sample basis and have access to the current model state as in HVP.

## G Learning View Generation

### G.1 Adversarial Learner

In this experiment, we explore adversarially learning a network to output the transformation matrix for view generation. We use Spatial Transformer Network (STN) Jaderberg et al. (2015) to allow generating views by producing 6D transformation matrices (allowing translating, rotating, shearing, scaling, affine transformations, and combinations thereof) in a differentiable way since most common augmentations are not off-the-shelf differentiable. We optimize the STN jointly with the DINO objective and a ViT-tiny. We train it alongside the actual pretrained network using the same (inverted) objective. For our experiments, we use DINO with multi-crop, i.e. 2 global and 8 local heads. As STN we use a small CNN followed by a linear layer for outputting the 10*6D transformation matrices. In this scenario, we use a ViT-tiny/16 with a 300 epoch pretraining on CIFAR10 with a batch size of 256. All other hyperparameters are identical to the ones reported in the DINO paper.

Figure 9 visualizes the procedure. The STN takes the raw image input and generates a number of transformation matrices that are applied to transform the image input into views. These views are then passed to the DINO training pipeline. Both networks are trained jointly with the same loss function. DINO is trained with its original contrastive objective, where the STN is trained by inverting the gradient after the DINO during backpropagation.

The STN, without using auxiliary losses, starts zooming in and generating single-color views. To counteract this behavior, we experimented with different penalties on the transformation matrices produced by the STN. For instance, in order to limit the zooming pattern, we can use the determinants of the sub-matrices of the transformation matrix to penalize based on the area calculated and apply a regression loss (e.g. MSE). We refer to this type of penalty as *Theta Crops Penalty* (TCP). Additionally, we also restrict its parameters to stay within a sphere with different parameters for local and global crops. Next to determinant-based penalty losses, we also experimented with other penalty functions such as the weighted MSE between the identity and the current transformation matrix or penalties based on histograms of the input image and generated views after applying the transformation. To avoid strong uni-dimensional scaling behavior, we also implemented restricting scaling in a symmetric

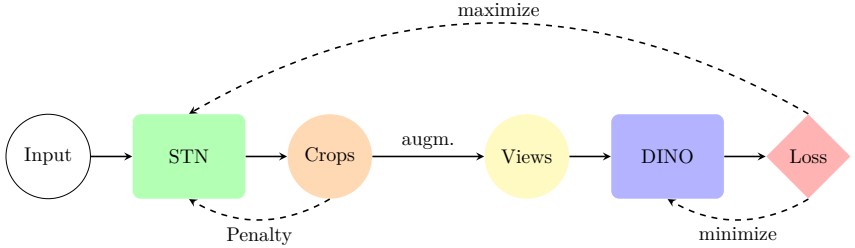

Figure 9: Illustration of adversarial learning with a Spatial Transformer Network (STN) jointly with contrastive learning (here: DINO).

way (i.e. applied to both x and y dimensions) and refer to this as *scale-sym.*. We report our best results in Table 5 which are all TCP-based. As can be seen, no setting is able to outperform the baseline. Our best score was achieved with translation-scale-symmetric which is very similar to random cropping. When removing the symmetries in scaling, the performance drops further. Removing a constraint adds one transformation parameter and therefore one dimension. This can be seen as giving more capacity to the adversarial learner which in turn can make the task significantly harder. Similarly, when adding rotation, the performance drops further and in part drastically. This is on the one hand due to the penalties not being fully able to restrict the output of the STN. On the other, the task of extracting useful information from two differently rotated crops is even harder, and learning spatial invariance becomes too challenging. All in all, we experienced two modes: either the STN is too restricted, leading to *static output* (i.e. independent of image content, the STN would produce constant transformation matrices) or the STN has too much freedom, resulting in extremely difficult tasks. See Fig. 10 for an example of the former behavior.

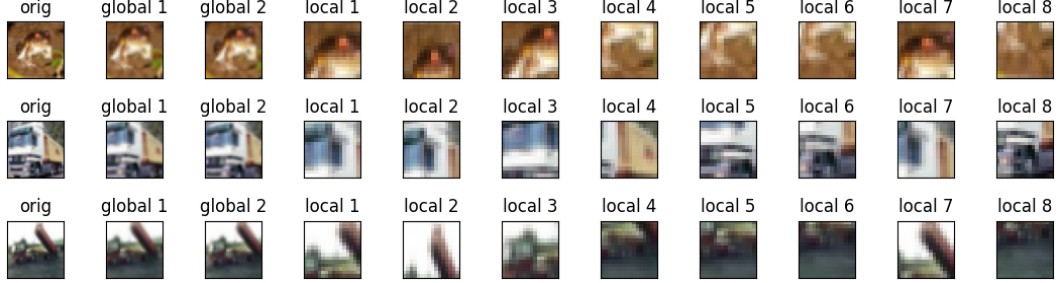

Figure 10: Example for static output behavior of the STN.

| Mode | Penalty | Lin. | F.T. |
|---|---|---|---|
| baseline | - | 86.1 | 92.7 |
| translation-scale-sym. | TCP | 83.7 | 90.3 |
| translation-scale | TCP | 82.8 | 89.7 |
| rotation-translation | TCP | 56.7 | - |
| rotation-translation-scale | TCP | 31.7 | - |
| rotation-translation-scale-sym. | TCP | 77.6 | - |
| affine | TCP | 78.3 | 83.5 |

Table 5: **Linear evaluation and finetuning classification performance on CIFAR10**. Top-1 accuracy on the validation set of CIFAR10 for our best results reported with different STN transformation modes.

## G.2 Cooperative Learner

To investigate the effect of a *cooperative*, i.e. easy pair selection, we conducted a small experiment. Instead of selecting the pair yielding the worst loss, we inverted the objective and selected the pair with the best loss. As expected, this led to model collapses with a linear eval. performance of 0.1%. This result is in line with previous findings that highlight the importance of strong augmentations in CL.

# H Additional Results

## H.1 Object Detection and Instance Segmentation

Here we provide more detailed results on object detection and instance segmentation, shown in Tab. 6 and Tab. 7 respectively. We followed iBOT's default configurations which employed Cascade Mask R-CNN as the task layer.

| Method | Arch. | Object Detection | | | Instance Segmentation | | |
|---|---|---|---|---|---|---|---|
| | | $AP^b$ | $AP^b_{50}$ | $AP^b_{75}$ | $AP^m$ | $AP^m_{50}$ | $AP^m_{75}$ |
| iBOT | ViT-S/16 | 47.00 | 66.13 | 50.63 | 40.67 | 63.10 | 43.37 |
| + HVP | ViT-S/16 | 47.27 | 66.50 | 50.90 | 40.90 | 63.50 | 43.83 |
| **Improvement** | | **+0.27** | **+0.37** | **+0.27** | **+0.23** | **+0.40** | **+0.47** |
| DINO | ViT-S/16 | 46.50 | 65.90 | 50.30 | 40.43 | 62.83 | 43.27 |
| + HVP | ViT-S/16 | 47.07 | 66.37 | 50.63 | 40.80 | 63.37 | 43.87 |
| **Improvement** | | **+0.57** | **+0.47** | **+0.33** | **+0.37** | **+0.53** | **+0.60** |

Table 6: Additional object detection and instance segmentation results on the COCO dataset. The ViT-S/16 models were pretrained for 100 epochs.

| Method | Arch. | Object Detection | | | Instance Segmentation | | |
|---|---|---|---|---|---|---|---|
| | | $AP^b$ | $AP^b_{50}$ | $AP^b_{75}$ | $AP^m$ | $AP^m_{50}$ | $AP^m_{75}$ |
| iBOT | ViT-S/16 | 47.60 | 66.80 | 51.33 | 41.10 | 63.63 | 44.07 |
| + HVP | ViT-S/16 | 48.03 | 67.13 | 51.73 | 41.50 | 64.23 | 44.40 |
| **Improvement** | | **+0.43** | **+0.33** | **+0.40** | **+0.40** | **+0.60** | **+0.33** |
| DINO | ViT-S/16 | 47.27 | 66.60 | 51.00 | 41.00 | 63.63 | 44.03 |
| + HVP | ViT-S/16 | 47.50 | 67.00 | 51.33 | 41.17 | 64.00 | 44.13 |
| **Improvement** | | **+0.23** | **+0.40** | **+0.33** | **+0.17** | **+0.37** | **+0.10** |

Table 7: Additional object detection and instance segmentation results on the COCO dataset. The ViT-S/16 models were pretrained for 300 epochs.

# I Hyperparameters

## I.1 Evaluations on ImageNet

### I.1.1 DINO

For DINO, we report the ViT pretraining hyperparameters in Table 8 (ViT-S) and Table 9 (ViT-B) which are the original ones as reported by the authors. Note, for HVP we limit the total number of comparisons to 128 across all heads. Linear evaluation is executed for 100 epochs and we use a learning rate of 0.00075, SGD optimizer (AdamW Loshchilov & Hutter (2019) during pretraining), a batch size of 1024, a momentum of 0.9, and no weight decay (as reported by the authors).

| Hyperparameter | Value | Hyperparameter | Value |
|---|---|---|---|
| architecture | vit-small | epochs: | 100 |
| img-size | 224 | warmup-epochs: | 10 |
| patch-size | 16 | freeze-last-layer: | 1 |
| out-dim | 65536 | lr: | 0.0005 |
| norm-last-layer | true | min-lr: | 1.0e-06 |
| momentum-teacher | 0.996 | optimizer: | AdamW |
| use-bn-in-head | false | weight-decay: | 0.04 |
| teacher-temp | 0.04 | weight-decay-end: | 0.4 |
| warmup-teacher-temp | 0.04 | global-crops-scale: | 0.4, 1.0 |
| warmup-teacher-temp-epochs | 0 | global-crops-size: | 224 |
| fp16 | true | local-crops-number: | 8 |
| batch-size | 512 | local-crops-scale | 0.05, 0.4 |
| clip-grad | 3.0 | local-crops-size: | 96 |
| drop-path-rate | 0.1 | | |

Table 8: Pretraining ImageNet hyperparameters for the runs with DINO ViT-S/16. For 300 epochs, we use a batch size of 1024.

| Hyperparameter | Value | Hyperparameter | Value |
|---|---|---|---|
| architecture | vit-base | epochs: | 400 |
| img-size | 224 | warmup-epochs: | 10 |
| patch-size | 16 | freeze-last-layer: | 3 |
| out-dim | 65536 | lr: | 0.00075 |
| norm-last-layer | true | min-lr: | 2.0e-06 |
| momentum-teacher | 0.996 | optimizer: | AdamW |
| use-bn-in-head | false | weight-decay: | 0.04 |
| teacher-temp | 0.07 | weight-decay-end: | 0.4 |
| warmup-teacher-temp | 0.04 | global-crops-scale: | 0.25, 1.0 |
| warmup-teacher-temp-epochs | 50 | global-crops-size: | 224 |
| fp16 | false | local-crops-number: | 10 |
| batch-size | 1024 | local-crops-scale: | 0.05, 0.25 |
| clip-grad | 0.3 | local-crops-size: | 96 |
| drop-path-rate | 0.1 | | |

Table 9: Pretraining ImageNet hyperparameters for the runs with DINO ViT-B/16.

### I.1.2 SIMSIAM

In Table 10, we report the ResNet-50 pretraining hyperparameters. Linear evaluation is executed for 90 epochs (as reported by the SimSiam authors) and we use a learning rate of 0.1, LARS optimizer You et al. (2017), a batch size of 4096, and no weight decay.

### I.1.3 SIMCLR

We report the ResNet-50 pretraining hyperparameters for SimCLR in Table 11. Linear evaluation is executed for 90 epochs with a learning rate of 0.1, SGD optimizer, batch size of 4096, and no weight decay.

### I.2 TRANSFER TO OTHER DATASETS AND TASKS

For linear evaluation on the transfer datasets, we simply used the same hyperparameters for linear evaluation on ImageNet (DINO and SimSiam respectively). For finetuning DINO ViT-S/16, we used the hyperparameters reported in Table 12 and for SimSiam ResNet-50 we used the hyperparameters in Table 13

| Hyperparameter | Value |
|---|---|
| architecture | resnet50 |
| batch-size | 512 |
| blur-prob | 0.5 |
| crops-scale | 0.2, 1.0 |
| crop-size | 224 |
| feature-dimension | 2048 |
| epochs | 100 |
| fix-pred-lr | true |
| lr | 0.05 |
| momentum | 0.9 |
| predictor-dimension | 512 |
| weight-decay | 0.0001 |
| optimizer | SGD |

Table 10: Pretraining ImageNet hyperparameters for the runs with SimSiam. For 300 epochs, we use a batch size of 1024.

| Hyperparameter | Value |
|---|---|
| architecture | resnet50 |
| proj-hidden-dim | 2048 |
| out-dim | 128 |
| use-bn-in-head | true |
| batch-size | 4096 |
| optim | LARS |
| lr | 0.3 |
| sqrt-lr | false |
| momentum | 0.9 |
| weight-decay | 1e-4 |
| epochs | 100 |
| warmup-epochs | 10 |
| zero-init-residual | true |

Table 11

| Hyperparameter | CIFAR10 | CIFAR100 | Flowers102 | iNat 21 | Food101 |
|---|---|---|---|---|---|
| lr | 7.5e-6 | 7.5e-6 | 5e-5 | 5e-5 | 5e-5 |
| weight-decay | 0.05 | 0.05 | 0.05 | 0.05 | 0.05 |
| optimizer | AdamW | AdamW | AdamW | AdamW | AdamW |
| epochs | 300 | 300 | 300 | 100 | 100 |
| batch-size | 512 | 512 | 512 | 512 | 512 |

Table 12: Finetuning hyperparameters for DINO ViT-S/16.

| Hyperparameter | CIFAR10 | CIFAR100 | Flowers102 | iNat 21 | Food101 |
|---|---|---|---|---|---|
| lr | 7.5e-6 | 5e-6 | 5e-4 | 7e-5 | 5e-6 |
| weight-decay | 0.05 | 0.05 | 0.05 | 0.05 | 0.05 |
| optimizer | AdamW | AdamW | AdamW | AdamW | AdamW |
| epochs | 300 | 300 | 300 | 100 | 100 |
| batch-size | 512 | 512 | 512 | 512 | 512 |

Table 13: Finetuning hyperparameters for SimSiam and ResNet-50.

## I.3   Object Detection and Instance Segmentation

Our experiments utilized Open MMLab's detection library Chen et al. (2019) for object detection and instance segmentation on COCO Lin et al. (2014). We followed iBOT's default configuration.

| Obj. Det. & Inst. Segm. on COCO | |
| --- | --- |
| Hyperparameter | Value |
| epochs | 12 |
| batch-size | 32 |
| lr | 0.02 |

Table 14: Hyperparameters object detection and instance segmentation on COCO.

## J   Computational Overhead of HVP

The additional forward passes that HVP introduces for the selection phase increase the time complexity of the individual baseline methods. Several possible approaches exist to mitigate this overhead, one of which is to alternate between the vanilla and HVP training step. We measured the overhead factors for different alternating frequencies (i.e., after how many training steps we use the hard views from HVP; we refer to this as *step*) for SimSiam, DINO, and iBOT which we report in Table 15. Below, we propose additional ways that may allow using hard views more efficiently.

| Step | SimSiam (RN50) | DINO (ViT-S/16) | DINO (RN50) | iBOT (ViT-S/16) |
| --- | --- | --- | --- | --- |
| 1 (HVP) | x1.64 | x2.21 | x2.01 | x2.13 |
| 2 | x1.38 | x1.61 | x1.56 | x1.57 |
| 3 | x1.32 | x1.43 | x1.42 | x1.38 |
| 4 | x1.29 | x1.34 | x1.35 | x1.29 |

Table 15: Slowdown factors for HVP and the alternating training method, where *step* refers to the interval at which HVP is applied during training (i.e., step=1 refers to full HVP and step=3 indicates that HVP is approx. used 33% of the training time). Measurements are averages across 3 seeds.

As an alternative to the alternating training, we also experimented with a 50% image resolution reduction for the selection phase but observed that the final performance was negatively affected by it or that baseline performance could not be improved.

The details of hardware and software used for this analysis are: one single compute node with 8 NVIDIA RTX 2080 Ti, AMD EPYC 7502 (32-Core Processor), 512GB RAM, Ubuntu 22.04.3 LTS, PyTorch 2.0.1, CUDA 11.8. For DINO's 2 global and 8 local views (default), applying HVP with nviews=2 sampled for each original view results in 4 global and 16 local views. Since considering all combinations would yield over 77k unique comparisons ($\binom{4}{2} \times \binom{16}{8}$), to remain tractable, we limit the number of total comparisons to 64. For training experiments that exceed the limit of 8x RTX 2080 Ti, we apply gradient accumulation.

While technically there can be a memory overhead with HVP, with the number of sampled views chosen in this paper, the backward pass of the methods that compute gradients only for the selected view pair still consumes more memory than the selection part of HVP (even for 8 sampled views in SimSiam). Note, that selection and the backward computation are never executed at the same time but sequentially.

We emphasize that further ways exist to optimize HVP's efficiency which remain to be explored. For instance:

- using embeddings of views from "earlier" layers in the networks or
- using 4/8 bit low-precision for the view selection or
- using one GPU just for creating embeddings and selecting the hardest views while the remaining GPUs are used for learning or
- other approaches to derive manual augmentation policies or
- bypassing forwarding of similar pairs.

# K    HARD VIEW PRETRAINING OBJECTIVES

## K.1    SIMCLR

In this section, we are going to introduce the application of HVP with the SimCLR objective. Assume a given set of images $\mathcal{D}$, an image augmentation distribution $\mathcal{T}$, a minibatch of $M$ images $\mathbf{x} = \{x_i\}_{i=1}^M$ sampled uniformly from $\mathcal{D}$, and two sets of randomly sampled image augmentations $A = \{t_i \sim \mathcal{T}\}_{i=1}^M$ and $B$ sampled from $\mathcal{T}$. We apply $A$ and $B$ to each image in $\mathbf{x}$ resulting in $\mathbf{x}^A$ and $\mathbf{x}^B$. Both augmented sets of views are subsequently projected into an embedding space with $\mathbf{z}^A = g_\theta(f_\theta(\mathbf{x}^A))$ and $\mathbf{z}^B = g_\theta(f_\theta(\mathbf{x}^B))$ where $f_\theta$ represents an encoder (or backbone) and $g_\theta$ a projector network. Contrastive learning algorithms then minimize the following objective function:

$$\mathcal{L}(\mathcal{T}, \mathbf{x}; \theta) = -\log \frac{\exp(\mathrm{sim}(\mathbf{z}_i^A, \mathbf{z}_i^B)/\tau)}{\sum_{i \neq j} \exp(\mathrm{sim}(\mathbf{z}_i^A, \mathbf{z}_j^B)/\tau)} \tag{3}$$

where $\tau$ denotes a temperature parameter and *sim* a similarity function that is often chosen as cosine similarity. Intuitively, when optimizing $\theta$, embeddings of two augmented views of the same image are attracted to each other while embeddings of different images are pushed further away from each other.

To further enhance the training process, we introduce a modification to the loss function where instead of having two sets of augmentations $A$ and $B$, we now have "N" sets of augmentations, denoted as $\mathcal{A} = \{A_1, A_2, \ldots, A_N\}$. Each set $A_i$ is sampled from the image augmentation distribution $\mathcal{T}$, and applied to each image in $\mathbf{x}$, resulting in "N" augmented sets of views $\mathbf{x}^{A_1}, \mathbf{x}^{A_2}, \ldots, \mathbf{x}^{A_N}$.

Similarly, we obtain $N$ sets of embeddings $\mathbf{z}^{A_1}, \mathbf{z}^{A_2}, \ldots, \mathbf{z}^{A_N}$ through the encoder and projector networks defined as:

$$\mathbf{z}^{A_i} = g_\theta(f_\theta(\mathbf{x}^{A_i})), \quad i = 1, 2, \ldots, N$$

We then define a new objective function that seeks to find the pair of augmented images that yield the highest loss. The modified loss function is defined as:

$$\mathcal{L}_{\max}(\mathcal{T}, \mathbf{x}; \theta) = \max_{k,l:k \neq l} \mathcal{L}(\mathcal{T}, \mathbf{x}; \theta)_{kl}$$

where

$$\mathcal{L}(\mathcal{T}, \mathbf{x}; \theta)_{kl} = -\log \frac{\exp(\mathrm{sim}(\mathbf{z}_k^{A_k}, \mathbf{z}_k^{A_l})/\tau)}{\sum_{i \neq j} \exp(\mathrm{sim}(\mathbf{z}_i^{A_k}, \mathbf{z}_j^{A_l})/\tau)}$$

and $k, l \in \{1, 2, \ldots, N\}$ and $i, j \in \{1, 2, \ldots, M\}$.

For each iteration, we evaluate all possible view pairs and contrast each view against every other example in the mini-batch. Intuitively, the pair that yields the highest loss is selected, which is the pair that at the same time minimizes the numerator and maximizes the denominator in the above equation. In other words, the hardest pair is the one, that has the lowest similarity with another augmented view of itself and the lowest dissimilarity with all other examples.

