# OpenReview forum: "Beyond Random Augmentations: Pretraining with Hard Views"
_ICLR.cc/2025/Conference — ICLR 2025 Poster_

### Official Review · Reviewer_k3VL · 2024-10-27

**Soundness:** 3
**Presentation:** 3
**Contribution:** 2
**Rating:** 6
**Confidence:** 3

**Summary:**

The paper introduces Hard View Pretraining (HVP), a novel self-supervised learning method aimed at improving pretraining pipelines by selecting hard views. Traditional SSL methods rely on random augmentations to create image views for model training. The authors hypothesize that by selecting the hardest views (those yielding higher loss), the learning process can be improved, resulting in better model performance. The method can be seamlessly combined with existing method like DIMO, SimSiam, iBOT and SimCLR, and the results show the effectiveness across many architectures and datasets.

**Strengths:**

- Good writing and clarity. The motivation is well-articulated, and the differences from prior work are clearly outlined. Overall, the writing is strong and clear.

- Simplicity. The approach is straightforward to implement and adaptable to a variety of architectures.

**Weaknesses:**

- The related works in Sec 2.2 are not comprehensive. There might be missing references for view selection. For example, Tian et al. [1] has a thorough discussion on the topic of "What Are the Optimal Views for Contrastive Learning?" Similar to your paper, Tian et al.  [1] studies the input view for contrastive learning. Moreover, Peng et al. [2] propose to generate contrastive views which could avoid most false positives (i.e., object vs. background). Similar to your paper, Peng et a. [2] studies the view selection. Thus, to clarity the distinction from the prior research, it is suggested to discuss the relation with the semantic-aware view selection in [2].

- Compuational overhead. The data augmentation significantly increases computational demands. Table 15 shows that HVP slows down the training of all models, despite efforts to use the hard view more efficiently. Please ensure a fair comparison under the same training budget or runtime. For example, the baseline (DINO) could be trained longer to match the computational budget of DINO+HVP.


- Minors. The authors report achieving a state-of-the-art 78.8% linear probing accuracy on ImageNet. To ensure a fair and comprehensive comparison, please include the results of DINO+HVP using the ViT-B/16 architecture (400 epochs) in the main table. Typo: the legend of figure 6 should be DINO+HVP rather than DINO+HVS.

- Minors. Lack of theoretical analysis: The method could be framed within the context of regularization techniques for self-supervised learning.

**Questions:**

- To leverage the computed embeddings from all views, one can take the average embedding and identify the one with the largest distance from it. This selected embedding can then be contrasted with either the other individual embeddings or their average. This might ensure that the additional computation from data augmentation is utilized effectively. Would adopting this strategy improve the method?

- In Appendix C, the training loss of DINO is notably high during the early stages. Could HVP be applied only in the later stages of training instead of throughout the entire process? If so, would it still enhance performance?

---

### Official Review · Reviewer_AeuA · 2024-11-02

**Soundness:** 3
**Presentation:** 3
**Contribution:** 2
**Rating:** 8
**Confidence:** 5

**Summary:**

The authors have proposed and validated an augmentation  method to train SSL models. The methods finds hardest views (HVP) based on loss in SSL to refine the learned representations. Thorough investigation has been done with DINO, SimSiam, SimCLR.
The method is simple and easy to incorporate in existing SSL pipelines. Transfer learning experiments have been done, many datasets are used.

**Strengths:**

The paper is well written, so many experiments have been done.
The method is simple and easy to incorporate in existing SSL pipelines, maybe seen as a plug-n-play method.

**Weaknesses:**

When we choose hardest views based on loss value, then it is certainly encouraging few augmentation strategies over others which defies the purpose of randomness of augmentations.
So, it may results performance improvement on unseen example (validation set) of the dataset ion which model is trained however, generalizability of model become questionable in reference to  domain adaptation.

Initially the model parameters are not effective, therefore, the higher loss may not be a good indicator of hard view.

As per Table 1, it is evident that longer pretraining is improving the performance even with original methods. Now, computation analysis suggests HVP approximately require 2x time of computation than original method (simSiam, simCLR, DINO iBOT), respectively. It conveys if original methods are pretrained twice number of epochs then they consume same computation resources like HVP. So, maybe, a fair comparison from computation perspective to compare HVP vs Original methods might require pretraining double. Like DINO with 100/300 epochs improves.

Random crop is dominant augmentation to receive the hard view as per loss, it introduces , i) information loss in visual concept, ii) spatial characteristics are randomly changed. Thus, it is important to understand whether hypothesis behind HVP stand without random crop.
The nonlinear transforms such as MPD and LCM two very recent augmentations without crop. It would be good to comment on such augmentations.
MPD: Möbius Transform for Mitigating Perspective Distortions in Representation Learning
LCM: Log Conformal Maps for Robust Representation Learning to Mitigate Perspective Distortion

**Questions:**

Follow strengths and weaknesses.

---

### Official Review · Reviewer_mBDk · 2024-11-03

**Soundness:** 3
**Presentation:** 3
**Contribution:** 3
**Rating:** 5
**Confidence:** 3

**Summary:**

The main content of this article is to introduce a new self supervised learning (SSL) pre training method called Hard View Pretraining (HVP). The core idea of this method is to enhance the learning performance of the model by selecting views with higher difficulty (i.e. views that generate higher loss). The HVP strategy includes the following iterative steps:

1. Randomly sample multiple views and propagate each view forward through a pre trained model.

2. Create a pair of two views and calculate their loss.

3. Adversarially select the view pairing that generates the highest loss based on the current model state.

4. Perform backpropagation on the selected view pairing.

In addition, the author also explores the computational cost of HVP, its integration with existing methods, and how to optimize the efficiency of HVP.

**Strengths:**

1. Innovative approach: A new self supervised learning pre training method HVP has been proposed, which improves the model's generalization ability by selecting difficult views. This is a novel research direction.

2. Wide applicability: The HVP method is not only applicable to one SSL method, but can be integrated into various popular SSL frameworks such as SimSiam, DINO, iBOT, and SimCLR, demonstrating good compatibility.

3. Significant performance improvement: HVP has shown better performance than existing methods on multiple datasets and tasks, particularly achieving new optimal accuracy on DINO ViT-B/16.

**Weaknesses:**

Although this article proposes a promising self supervised learning pre training method HVP and demonstrates its effectiveness on multiple tasks, there are also some potential shortcomings:

1. Computational cost: The HVP method requires additional forward propagation to select the most difficult view pairs, which may increase the computational cost of training, especially on large-scale datasets and complex models. Please try to compare the computational cost of proposed approach and existing ones for a more comprehensive performance evaluation.

2. Hyperparameter adjustment: Although HVP does not require adjusting too many hyperparameters, some adjustments may still be needed in determining the number of views and selecting the most difficult view pairs, which may require additional experimental and computational resources.

3. Validation of generalization ability: Although the article demonstrates the effectiveness of HVP on multiple datasets and tasks, further validation is needed for its generalization ability on a wider range of tasks and datasets.

4. Theoretical analysis: The article mainly focuses on experimental results, and the theoretical analysis behind why HVP is effective may not be in-depth enough, especially in understanding how the model learns useful features from difficult views. Maybe a proof of underlying theory (e.g., how HVP affects the geometry of the learned feature space, or providing theoretical bounds on the expected improvement from using hard views) will be helpful.

5. Memory overhead: In some cases, HVP may increase memory overhead, especially when dealing with a large number of views and complex models, which may limit its application in resource constrained environments.

6. Stability of opponent models: When exploring the capacity of opponent models, the article mentions the issue of algorithm stability, indicating that in some cases, HVP may be affected by model crashes. Please conduct a systematic study of how different adversarial strengths affect training stability and performance.

7. Dependence on existing processes: Although HVP can be integrated into existing SSL frameworks, its effectiveness may depend on specific data augmentation distributions and model architectures, which may limit its applicability in different settings.

8. Diversity of experimental design: Although the article provides a wide range of experiments to validate the effectiveness of HVP, more experiments may be needed to explore the performance of HVP under different conditions, such as different learning rates, batch sizes, and training period.

**Questions:**

Please refer to weakness for the details.

---

### Official Review · Reviewer_xunC · 2024-11-03

**Soundness:** 2
**Presentation:** 3
**Contribution:** 2
**Rating:** 5
**Confidence:** 4

**Summary:**

This paper proposes Hard View Pretraining (HVP), an approach for improving self-supervised learning (SSL) by selecting challenging, high-loss views during pretraining. Unlike conventional SSL methods that rely on random augmentations, HVP samples multiple views for each input, computes the pairwise loss, and selects the view pair with the highest loss. This adversarial selection of views enables the model to learn more robust representations by iteratively exposing it to more difficult training examples. The method integrates seamlessly with popular SSL frameworks, including DINO, SimSiam, iBOT, and SimCLR, demonstrating consistent performance gains across various models and transfer tasks. HVP achieves state-of-the-art results on the DINO ViT-B/16 model with a 0.6% improvement in linear evaluation accuracy (78.8%) and demonstrates its scalability across different architectures and datasets.

**Strengths:**

- HVP introduces a straightforward, loss-based hard view selection mechanism that enhances SSL training without requiring additional components or extensive hyperparameter tuning. This simplicity makes it highly practical for integration into existing SSL pipelines.
- The paper presents extensive experiments across different SSL frameworks, model architectures (e.g., CNNs and Vision Transformers), and datasets. This broad evaluation supports the generalizability of HVP and its effectiveness in improving SSL methods.
- The method is demonstrated on large-scale datasets such as ImageNet and COCO, where it consistently outperforms baselines, particularly in challenging settings. HVP’s strong performance on both linear evaluation and transfer tasks, including object detection and segmentation, highlights its robustness and adaptability to diverse downstream applications.

**Weaknesses:**

1. HVP’s reliance on high-loss pair selection may result in false positive pairs (i.e., views from different instances within the same image) being chosen, which could hinder representation learning. The paper does not clearly address whether the current HVP method can effectively avoid or mitigate this issue.
2. The related work section does not thoroughly discuss other existing view construction methods such as [1,2,3,4] nor does it compare HVP with these methods experimentally. The absence of experimental comparisons with these methods limits the paper’s ability to demonstrate HVP’s distinct advantages in SSL.
3. HVP’s reliance on loss maximization may limit its effectiveness in tasks where pairwise loss is not a straightforward metric, such as pixel-level reconstruction tasks (e.g., Masked Autoencoders[5]) or relational consistency tasks like Relational Knowledge Distillation[6]. The paper could explore adaptations to broaden HVP’s applicability in such contexts.

[1]Tamkin, Alex, et al. “Viewmaker networks: Learning views for unsupervised representation learning.” CVPR 2020.

[2] Peng, Xiangyu, et al. "Crafting better contrastive views for siamese representation learning." CVPR 2022.

[3] Han, Ligong, et al. "Constructive assimilation: Boosting contrastive learning performance through view generation strategies." arXiv:2304.00601.

[4]Li, Xiaojie, et al. “GenView: Enhancing View Quality with Pretrained Generative Model for Self-Supervised Learning.” ECCV. 2024.

[5] He, Kaiming, et al. “Masked autoencoders are scalable vision learners.” CVPR. 2022.

[5]Zheng, Mingkai, et al. “Weak Augmentation Guided Relational Self-Supervised Learning.” TPAMI 2024.

**Questions:**

1. How does HVP address the potential issue of false pairs in hard view selection? Were any additional measures considered for identifying or filtering these pairs?
2. Could the authors further clarify HVP’s unique advantages compared to existing view-construction methods and provide relevant experimental comparisons?
3. Given HVP’s reliance on loss maximization, which may limit its use in tasks with complex pairwise losses (e.g., MAE or relational consistency tasks like Relational KD), could the authors discuss potential adaptations to extend HVP’s applicability?

---

### Author Response · Authors · 2024-12-03
**Overview of Rebuttal Phase**

Dear Area Chairs,

We appreciate the reviewers' thoughtful evaluations of our paper and have carefully addressed their concerns. Below, we provide a unified discussion of the main issues raised and our responses, followed by the outcomes for each reviewer.

------
### Main Concerns and Our Responses

**Preliminary Note on Reviewer mBDk**

As previously noted, we believe Reviewer mBDk's review was generated by an LLM, as they contain generic phrasing and lack specific engagement with our work. The reviewer did not engage in any discussion during the rebuttal phase, and we received no sign of life from them. Despite this, we addressed and refuted all their points in good faith, providing detailed responses to their critique but again did not receive a response.

- Broader Applicability and Experimental Comparisons
    - Concern: Broader applicability of HVP and the sufficiency of experimental comparisons with baselines.
    - Response: We demonstrated that our method is broadly applicable to other domains and objectives, as evidenced by the iBOT result incorporating a Masked Image Modeling (MIM) objective, as well as additional baselines like DINO, SimCLR, and SimSiam, each of which emphasizes different hyperparameter configurations as well as different architectures. Compared to related work, our method is among the few to conduct extensive evaluations on full-scale ImageNet. Many prior approaches either omit ImageNet entirely or focus only on reduced subsets, whereas we present results on a variety of tasks and datasets, including ImageNet and downstream tasks.

- Fair Comparisons Under Equivalent Computational Budgets
    - Concern: The computational overhead of HVP and the need for fair comparisons with baselines under similar budgets.
    - Response: Like prior impactful works such as DINO and multi-crop, we followed the path of prioritizing achieving state-of-the-art (SOTA) results over an efficient method with similar computational complexity as the baseline. With our limited computational setup, we already demonstrated through extensive experiments that HVP improves upon baselines consistently and robustly across both longer and shorter runs. The consistent track record across diverse setups gives us strong confidence in HVP's robustness and its potential to yield further advancements.

- Discussion of Related Work
    - Concern: Insufficient discussion of related methods, particularly Han et al., Li et al., Tian et al., and Peng et al.
    - Response: We contrasted and positioned HVP against all mentioned related works. They mostly rely on learning-based methods that require training additional auxiliary or adversarial networks, which adds significant complexity and overhead to existing pipelines. In contrast, HVP avoids this by being a lightweight, learning-free approach that leverages the current model state. Moreover, as mentioned, many of these works do not train on full ImageNet. Upon acceptance, we will include the suggested references and expand the related work section to address these comments.

-----

### Per-Reviewer Outcomes
- Reviewer AeuA: Increased their score from 6 to 8, fully accepting our responses, recognizing the robustness and novelty of our approach, as well as the breadth of our experimental evaluation.
- Reviewer xunC: Increased their score from 3 to 5 after engaging with our responses and acknowledging our clarifications.
- Reviewer k3VL: Increased their score from 5 to 6 after our rebuttal, engaging constructively and encouraging further exploration of computational comparisons in future work. To the best of our knowledge, we addressed all their critique regarding broad applicability.
- Reviewer mBDk: Did not engage during the rebuttal phase, but we addressed all their points comprehensively.

----

### Summary

We hope and believe that our rebuttal has addressed the majority of the concerns raised. Our work demonstrates a lightweight and effective approach to SSL, with robust results across 5 downstream datasets for finetuning and linear evaluation, object detection and segmentation, and the integration with the DINO, SimCLR, SimSiam, and iBOT frameworks trained and evaluated with ViTs and ResNets.

We kindly ask the Area Chairs to consider our responses and the improvements made during the rebuttal phase.

Thank you for your consideration and efforts.

Best regards.

---

### Meta-Review · Area_Chair_KKnN · 2024-12-21

**Metareview:**

The rebuttal provided clarifications about the proposed method and its analysis that were useful for assessing the paper's contribution and responded adequately to most reviewer concerns. After discussion, reviewer AeuA recommended acceptance, k3VL recommended marginal acceptance, xunC recommended marginally below acceptance. Reviewer mBDk was not involved in the rebuttal. The AC agreed this work is valuable to the ICLR community. The final version should include all reviewer comments, suggestions, and additional clarifications from the rebuttal.

**Additional Comments On Reviewer Discussion:**

NA

---

### Decision · Program_Chairs · 2025-01-22

Accept (Poster)